# *Betapapillomaviruses* in p16-Negative Vulvar Intraepithelial Lesions Associated with Squamous Cell Carcinoma

**DOI:** 10.3390/v15091950

**Published:** 2023-09-19

**Authors:** Taja Lozar, Aysenur Keske, Racheal S. Dube Mandishora, Qiqi Yu, Adam Bailey, Jin Xu, Massimo Tommasino, Stephanie M. McGregor, Paul F. Lambert, Tarik Gheit, Megan B. Fitzpatrick

**Affiliations:** 1McArdle Laboratory for Cancer Research, University of Wisconsin School of Medicine and Public Health, Madison, WI 53705, USA; tlozar@wisc.edu (T.L.);; 2University of Wisconsin Carbone Cancer Center, Madison, WI 53705, USA; 3University of Ljubljana, 1000 Ljubljana, Slovenia; 4Department of Pathology and Laboratory Medicine, University of Wisconsin School of Medicine and Public Health, Madison, WI 53726, USA; 5Center for Immunization and Infection Research in Cancer (CIIRC), Moffit Cancer Center, Tampa, FL 33612, USA; 6Medical Microbiology Unit, University of Zimbabwe Faculty of Health Sciences, Harare P.O. Box A178, Zimbabwe; 7Department of Pathology, Johns Hopkins School of Medicine, Baltimore, MD 21205, USA; qyu17@jhmi.edu; 8IRCCS Istituto Tumori Giovanni Paolo ll, 70124 Bari, Italy; 9International Agency for Research on Cancer, 69007 Lyon, France

**Keywords:** beta human papillomavirus, vulvar cancer, vulvar precancer

## Abstract

Approximately 40% of vulvar squamous cell carcinoma (vSCC) cases are etiologically associated with high-risk human papillomaviruses (HPVs) of the alpha genera (α-HPV) that cause other anogenital cancers; however, the etiology of α-HPV-negative vSCC is poorly understood. HPVs of the beta genera (β-HPV) are risk factors for cutaneous squamous cell carcinoma (cSCC) and may be related to carcinomas originating in other cutaneous sites such as the vulva. In this study, we investigate the presence of β-HPVs, with an emphasis on p16-negative squamous lesions adjacent to vSCC. We subjected 28 vulvar squamous intraepithelial lesions adjacent to vSCC for comprehensive HPV genotyping, p16 and p53 immunohistochemistry, and consensus morphology review. Selected cases were subjected to qPCR and RNA in situ hybridization. Clinical data were obtained from medical records. β-HPV DNA was detected in eight of ten p16-negative lesions and three of fourteen p16-positive high-grade squamous intraepithelial lesions. The HPV DNA loads in vulvar squamous intraepithelial lesions ranged between less than 1 HPV DNA copy per cell to more than 100 HPV DNA copies per cell. This is, to the best of our knowledge, the first report of the association of p16-negative vulvar intraepithelial squamous lesions with detection of β-HPVs. These findings expand possible etiologic mechanisms that may contribute to p16-negative lesions of the vulva.

## 1. Introduction

The most recent 2020 WHO classification of female genital tumors [1] classified vulvar squamous intraepithelial lesions into categories including high-risk human papillomavirus (HPV)-associated and HPV-independent lesions [2,3,4,5]. Typically, these lesions are categorized based on morphology with adjunct immunohistochemistry for p16 and p53; however, approximately 10% of squamous intraepithelial lesions do not fall into either category [2,3,6,7,8,9,10]. A new category of HPV-independent squamous intraepithelial lesions has also been described which includes vulvar aberrant maturation (VAM), differentiated exophytic intraepithelial lesion (deVIL), and vulvar acanthosis with altered differentiation [3,5,7,11,12,13,14]. These well-differentiated squamous lesions lack expression of p16 with either p53 wild-type or basal expression by immunohistochemistry and show an exophytic and/or verruciform growth pattern [15]. These lesions have been proposed as HPV-independent p53 wild-type verruciform acanthotic vulvar intraepithelial neoplasia (HPVi (p53wt) vaVIN) or HPV independent p53 abnormal (HPV-I p53abn), encompassing lesions that fall within a spectrum of differentiated exophytic vulvar intraepithelial lesions (deVIL) and vulvar acanthosis with altered differentiation (VAAD) [2,5,11,15,16,17,18]; however, no clear etiology has been identified.

Two hundred and twenty-nine (229) HPV genotypes are currently listed in the international HPV reference center list [19], with the majority belonging to the *Alphapapillomavirus (*α-HPV)*, Betapapillomavirus* (β-HPV), and *Gammapapillomavirus* (γ-HPV) genera, categorized as mucosal or cutaneous types based on their epithelial tropism [20]. Twelve mucosal HPVs belonging to the α-HPV genus have been associated with development of malignancies, and therefore have been classified as high risk by the World Health Organization’s International Agency on Research on Cancer (WHO/IARC) [20,21]. In contrast, a role for β-HPV in the initiation of carcinogenesis in cutaneous squamous cell carcinoma has been proposed, but there are insufficient causal data to classify them as definitively oncogenic [22]. Given the prevalence of β-HPV in multiple benign mucocutaneous sites, it is thought that these HPV types require cofactors to induce carcinogenesis, such as ultraviolet (UV) radiation and immunosuppression [23,24,25]. The most classic example of β-HPV-related cutaneous cancers occurs in patients with epidermodysplasia verruciformis (EV), in which HPV 5 and 8 have been identified in 90% of cSCCs [23,24,26,27]. Similar to EV, β-HPV types represent the majority of HPV infections in organ transplant recipients, where the rate of HPV detection in cSCC samples ranges from 70 to 90% [28]. Cutaneous squamous lesions associated with β-HPV can have APOBEC3 pathway mutations (*PIK3CA*) and *NOTCH1* mutations similar to those recently described in verruciform lesions of the vulva [11,13,14,24,29,30,31].

Since the vulva is a predominately cutaneous site with hair follicles involving the labia majora, we hypothesized that cutaneous-type (β- and/or γ-papillomaviruses) may also play a role in the carcinogenic initiation of vulvar lesions, as proposed in a subset of cutaneous squamous intraepithelial lesions [23,32,33]. In the vulva, cutaneous type HPVs have only been investigated in squamous cell carcinomas, where it is rarely identified and almost never exists as a single infection [34]. However, the absence of HPV in such frankly malignant lesions may be due to a role in the initiation of carcinogenesis, but not in its progression. Indeed, the viral load of HPV is higher in actinic keratoses than in invasive cSCC [35,36], and in transgenic mouse models, HPV is present in precursor intraepithelial lesions but no longer present in the resultant carcinomas after accumulation of mutations in oncogenic driver genes [37]. Typically, these models of HPV-induced cutaneous squamous lesions require concomitant UV radiation or alteration of the cellular response to UV stress [38]. 

Recent reports [39,40] on the lack of progress in the early diagnosis of vSCC outline the need for a better understanding of the development of vSCC and early prevention and detection strategies. Accurate identification of vulvar intraepithelial lesions is currently limited, in part, by the lack of accurate biomarkers, further necessitating the distinction and further study of the behaviors of these lesions. In addition, lesions with co-infections of β- and α-HPV types show more differentiation than those with α alone, which may have implications for emerging therapy (e.g., immunotherapy and therapeutic vaccinations) for these lesions, further highlighting the need for additional investigations of these categories. In this study, we analyzed a case series of vulvar squamous intraepithelial lesions associated with invasive squamous cell carcinoma for the presence of HPV types from genus α (*n* = 21), β (*n* = 46), and γ (*n* = 52), combined with immunohistochemistry (IHC) for p16 and p53, and a consensus morphology review. In addition, we performed quantitative PCR (qPCR) to determine the viral load for selected β-HPV types.

## 2. Materials and Methods

### 2.1. Case Selection and Interpretation

The inclusion criteria for this study were as follows: vulvar squamous cell carcinoma that was treated at the University of Wisconsin-Madison between 2009 and 2020, matched precursor and invasive block available, sufficient tissue available, and equal representation of p16-positive vs. -negative lesions. We performed a natural language search within the laboratory information system of the UW Health Department of Pathology and Laboratory Medicine, using the following search terms to enrich for p16-negative lesions: “keratinizing squamous cell carcinoma”, “well-differentiated squamous cell carcinoma”, “verrucous”, “exophytic”, “dVIN”, “differentiated”, “atypical squamous proliferation/lesion”, as well as a cohort of selected comparators including “squamous cell carcinoma”, “high-grade squamous intraepithelial lesion”, and “VIN”. The search revealed 93 cases from 2009 to 2020, of which 29 cases had a diagnosis of vSCC with adjacent identifiable intraepithelial lesions and formalin-fixed paraffin-embedded (FFPE) blocks with sufficient tissue available for HPV typing. Additional cases were excluded from the analysis that were categorized as condyloma (*n* = 1), or poorly defined atypical lesions not diagnostic of any intraepithelial lesion (*n* = 2). Finally, 28 cases with well-defined vulvar squamous intraepithelial lesions that were confirmed to be adjacent to vSCC, from 26 patients, were included in the analysis. 

For each case, the paraffin block containing the precursor intraepithelial lesion, but not invasive carcinoma, was selected for downstream analyses. The histology for each case was reviewed and reclassified based on a consensus review by 4 gynecologic pathologists, 3 of whom were blinded to the results and clinical history. A combined morphologic review in conjunction with p16 and p53 interpretation was used to classify the lesions, including independent reviews by two gynecologic pathologists (JX/CF). Lesions were categorized based on the most recent WHO 2020 criteria [41], i.e., the two main morphological patterns of high-grade squamous intraepithelial lesion (HSIL), including basaloid and warty, were included in HSIL. Categorization as dVIN was based on the presence of basal nuclear atypia, atypical mitosis, dyskeratosis, and anastomosing rete ridges. Lastly, verrucous exophytic lesions with acanthosis without the features of dVIN or HSIL were categorized as atypical squamous proliferation (ASP), which included lesions previously described as deVIL and VAAD [3,5,11,15,17,42]. 

### 2.2. Immunohistochemistry

When available, p16 and p53 immunohistochemical staining originally obtained during the work-up of the squamous lesions were reviewed. If p53 and p16 staining had not been performed at the time of diagnosis, cases were stained in the Translational Research Initiatives in Pathology (TRIP) Laboratory in the Department of Pathology and Laboratory Medicine. The clones used for the immunohistochemistry were the anti-p53 clone DO-7 (mouse monoclonal antibody 790-2912 Ventana Medical systems) or anti-p16 clone E6H4 (CINtec Histology, mouse monoclonal antibody, 705-4793 Ventana Medical Systems), performed on the Ventana Discovery Ultra BioMarker Platform (Roche, Indianapolis, IN, USA). A similar protocol was followed for each antibody (Appendix A). 

### 2.3. Immunohistochemistry Interpretation

IHC for p16 was considered to be positive if strong, block-positive, continuous staining across the cytoplasm and nuclei in at least the basal third of the epithelium was observed, and negative if there was focal, noncontiguous, or absent staining [9,43]. IHC for p53 was interpreted as described by others into six staining patterns: basal sparing, wild-type/weak scattered, basal overexpression, basal/parabasal overexpression, null/absent, and cytoplasmic staining, as previously described [18,44,45]. Stains were interpreted by four experienced gynecologic pathologists (M.B.F., J.X., S.M.M., and C.F.).

### 2.4. HPV Genotyping

The FFPE samples were sectioned in duplicate (3 × 10 μm from each block). At the International Agency for Research on Cancer (IARC, Lyon, France), DNA was prepared as previously described [27,46,47] and tested for the presence of α-, β- and γ-HPVs, using type-specific PCR bead-based multiplex genotyping assays that combine multiplex PCR and Luminex technology (Luminex, Austin, TX, USA), as described elsewhere [27,47]. Details are presented in Appendix A. 

### 2.5. Quantitative PCR

Quantitative PCR was carried out on samples that were positive for at least one of the β-HPV types and had sufficient FFPE tissue remaining for molecular analyses (at least 10 × 7 μm sections). For each case that was positive for at least one β-HPV type, two paraffin blocks were collected and analyzed, i.e., one block containing the intraepithelial lesion without the invasive carcinoma component, and one block containing the invasive carcinoma. DNA was isolated from the FFPE tissue using a QIAamp DNeasy Blood and Tissue Kit (Qiagen), according to the manufacturer’s instruction. Following isolation, the DNA concentration was measured using a Qubit fluorimeter. Type-specific qPCR protocols for HPV types 5, 9, 24, and 111 were developed at the UW-Madison and performed using an ABI 7900HT real-time PCR system (Applied Biosystems, Waltham, MA, USA) using TaqMan Gene Expression master mix (Thermo Fisher Scientific, Watham, MA, USA). HPV DNA copy numbers were determined using standard curves, generated in the same PCR run with dilutions of plasmids containing HPV genomic or amplicon sequences ranging from 10 to 106 copies per sample, as well as a beta globin standard curve using TaqMan Control Human Genomic DNA (Thermo Fisher Scientific). A no-template control using diH20 was included in each run. All standards were run in triplicates, while clinical samples were run in duplicate due to low sample volume. Each viral load assay contained HPV type-specific primers and probes that bind a region of the major capsid protein encoding L1 gene (Appendix A). Standard cycling conditions were used: 50 °C for 2 min, 95 °C for 10 min, and 50 cycles of 95 °C for 15 s and 60 °C for 1 min. Briefly, up to 8.5 μL total DNA isolate (calculated to include 3000 beta-globin gene copies of each sample as recommended by Weissenborn [48]) was suspended in a 11.5 μL reaction mixture containing 10.5 μL of 1× TaqMan Gene Expression Master Mix (Thermo Fisher Scientific), 400 nM forward primer, 400 nM reverse primer, and 100 nM probe. The amount of HPV genomic copies per reaction was corrected for cellular content with a beta globin qPCR to obtain the viral load, which is defined as copies per cell (c/cell). Detailed calculations are presented in Appendix A. The single-copy gene beta globin was quantified using beta globin primers and probe mix Hs00747223_g1 (Thermo Fisher Scientific). An additional plate using TaqMan Control Human Genomic DNA (Thermo Fisher Scientific) was run to determine unspecific binding of all HPV primers and probes, including no template controls. All analyses were conducted at the UW-Madison. The genotype-specific amplicons were provided by IARC. Each limit of detection (LOD) was calculated using the formula 3.3(σ/S), where σ is the standard deviation of the y-intercept, and S is the magnitude of the qPCR curve slope. Ordinary least squares regression of mean concentration on log10-transformed copy number was used to estimate σ and S.

### 2.6. RNA In Situ Hybridization (ISH)

RNA ISH was performed on all p16-negative samples with sufficient tissue available for analysis. ISH for HPV E6/E7 transcript was completed using RNAscope (2.5 HD Reagent Kit-Brown, 322300, Advanced Cell Diagnostics, Newark, CA, USA) with probes specific for 18 high-risk HPV genotypes (probe 312591), according to the manufacturer’s instructions. A negative and positive control slide was included with each run. In addition, RNA ISH was attempted for beta HPV genotypes 5, 9, 24, and 111 on samples selected for qPCR; however, this was technically challenging due to lack of positive controls (details are presented in Appendix A).

### 2.7. Statistical Analysis

We calculated viral load by dividing the number of copies of viral DNA detected by the total number of cells (estimated by the number of beta-globin copies divided by 2) in the sample. The patient’s categorical characteristics were presented as frequencies and proportions. Age was presented as mean and range. Student’s paired *t*-test, one-way ANOVA, log-rank test, and Fisher’s exact test were used for statistical comparisons. A *p*-value < 0.05 was considered to be significant. Statistical analysis was performed using SPSS v.24.0 (IBM Corp.) and GraphPad Prism v 10.02.

## 3. Results

### 3.1. Case Characteristics

Patient characteristics are presented in Table 1 and Appendix A. Altogether, 28 cases from 26 patients were included in the study. The mean and median ages were 62.2 years and 61.8 years, respectively. Most patients (25/26, 96%) were white. Three patients (3/26, 11.5%) had a history of immunosuppression or weakened immune function, i.e., two patients had a kidney transplant, and one patient had poorly controlled diabetes mellitus. 

Initial pathologic diagnoses were obtained for all cases. Of the 28 cases from 26 patients, two (7.1%) adjacent intraepithelial cases were initially classified as ASP, eight (28.6%) cases were classified as dVIN, six (21.4%) cases were classified as classic type HSIL, six (21.4%) cases were classified as classic/basaloid/warty vulvar intraepithelial neoplasia 3 (VIN3), three (10.7%) cases were classified as mixed classic/differentiated VIN3, one case was classified as verrucous/dVIN, and one case was classified as a verrucous squamous lesion. Altogether, eight (8/26, 30.8%) patients had p16 staining performed at diagnosis and eight (8/26, 30.8%) patients had p53 staining performed at diagnosis. Of those, two (2/26, 7.7%) patients had both p16 and p53 staining performed at diagnosis. For one case, a definitive intraepithelial diagnosis was not mentioned in the initial pathology report, and another case was described as neither fitting HPV-related, basaloid-type VIN3, nor dVIN.

After consensus morphology review supplemented with stains as necessary, cases were reclassified as dVIN (3/28, 10.7%), HSIL (15/28, 53.6%), and ASP (10/28, 35.7%). An interobserver variability analysis based on an initial independent morphology review revealed disagreements in three (3/9, 33.3%) cases of ASP, one (1/3, 33.3%) case of dVIN, and four (4/14, 28.6%) cases of HSILs (Appendix A). Detailed clinicopathologic data are provided in Table 1. Briefly, 15 (53.6%) cases were positive for block-like p16 staining and 13 (46.4%) cases were p16 negative. The mean age at diagnosis of most advanced documented lesion did not differ significantly between the morphology groups (*p* = 0.278) or between the p16-positive and -negative groups (*p* = 0.053). All three patients with a history of immunosuppression were p16 positive, however, this was not statistically significant (*p* = 0.23).

Of the two patients that had two different cases included in the study, one patient was diagnosed with HSIL (p16 positive, p53 BS) on both samples, and one patient was diagnosed with ASP (p16 negative, p53 OB) in both. A single case per patient was selected for downstream molecular analyses. Of the p16-negative cases (*n* = 13 cases from 12 patients), nine cases were considered to meet the morphological criteria to be considered ASPs (deVIL or VAAD). Most p16-positive cases were classified as classic/usual type high-grade squamous intraepithelial lesions (*n* = 15, HSIL). 

### 3.2. HPV Genotyping

Only samples with adequate DNA were included (*n* = 26 patients, 28 cases). Of these, eleven (11/26, 42.3%) patients had detectable β-HPV DNA by Luminex technology, and of those, three (3/26, 11.5%) patients as single genus and eight (8/26, 30.8%) patients as mixed β- and α-HPV infection (Table 2). One patient had detectable γ-HPV DNA detected as a co-infection with high-risk α-HPV and β-HPV. Overall, 18 patients had an α-HPV DNA detected, of those 12 patients were p16 positive, and 6 patients were p16 negative. The presence of β-HPV DNA differed significantly between the morphology categories, such that ASP morphology was significantly associated with β-HPV DNA (*p* = 0.014, Figure 1) particularly when detected in the absence of α-HPV. Most of the lesions that had a β-HPV detected showed either overexpressed basal or wild-type expression of p53. When detected in HSIL lesions, the morphology was most fitting with a well-differentiated, keratinizing and warty type, previously referred to as VIN3-warty type. The relationship of morphology and the presence of β-HPV is presented in Table 2 and in Figure 2.

#### 3.2.1. p16-Negative Lesions

Of the squamous intraepithelial lesions considered to be ASP, seven of ten (70%) cases had detectable β papillomavirus DNA (two cases had β-HPV only, five cases had mixed high-risk α and β types, and no cases had detectable low-risk α infection). Altogether, six of seven (85.7%) cases had a co-infection with at least one α papillomavirus type. All cases were p16 negative and, therefore, considered that high-risk HPV was not integrated based on the expression of surrogate marker (41). Of the three ASP lesions with detectable single genus β-HPV, two lesions had viral loads >1 c/cell.

Among the three cases that were considered to be dVIN, none showed p16 expression, and all showed either overexpressed basal or overexpressed basal/parabasal p53 by immunohistochemistry. One case showed overlapping features of dVIN and HSIL with HPV 24 detected in the intraepithelial lesion and was classified as dVIN with HSIL features based on the negative p16 and overexpressed p53 by immunohistochemistry [3,6].

#### 3.2.2. p16-Positive Lesions

Altogether, we identified 14 HSIL lesions (all p16 positive, 13/14 p53 basal sparing). Among the HSIL lesions, HPV 16 infections were the predominant infection (10/14, 71.4%, Table 2), and multiple infections with α-HPV and β- or γ-HPV were detected in three cases (3/14, 21.4%, one patient with transplant-related immunosuppression). 

### 3.3. HPV Viral Loads

After identifying β-HPV in selected intraepithelial lesions, we wanted to quantify viral copy number per cell, particularly in the single genus DNA, when compared with those harboring α-HPV as co-infections. Therefore, qPCR was performed on all cases where we had previously identified selected β-HPV genotypes by Luminex technology with sufficient tissue available in both the intraepithelial and invasive cancer tissue blocks. We identified nine cases with sufficient tissue left on both blocks, and blocks that contained only intraepithelial lesion were compared to those blocks with invasive carcinoma. Of those, five cases had sufficient DNA isolated from the intraepithelial block and three cases had sufficient DNA isolated from the invasive block to ensure adequate assay sensitivity. To determine HPV DNA copy numbers and input cell equivalents, type-specific qPCR protocols for selected β-HPV types were performed, i.e., HPV type 5, 9, 24, and 111 (Table 3).

The HPV DNA loads in vulvar intraepithelial biopsies ranged between less than 1 HPV DNA copy per cell to more than 100 HPV DNA copies per cell. After assessing LOD for each assay, we found that only HPV 9 and HPV 111 had significant viral copy numbers, i.e., significantly above LOD. For Patient 5, DNA from two separate intraepithelial biopsies was available for HPV genotyping and qPCR; one patient had detectable HPV 5 (with DNA load of 1.6 c/cell which was below the LOD, and therefore not considered significant), and the other patient had detectable HPV 111 (153 c/cell), with no detectable α-HPV in any of these lesions. In three patients, viral loads could be calculated and compared between the intraepithelial and invasive tissue block. The viral loads in invasive cancer blocks were lower than those in matched intraepithelial lesions in all three cases; however, these findings were considered to be significant only in the case of HPV 111. In addition, two patients (2/5, 40%) had co-infections of β-HPV with high-risk HPV 16. In Patient 50, the β-HPV viral loads were <1 c/cell in the intraepithelial lesion. In one patient with a single detectable genotype HPV 111, the β-HPV viral load was over 100 c/cell in both the intraepithelial lesion and invasive lesion. Viral type-specific β-HPV DNA detected in HPV-negative control human DNA ranged from 0.00003 (HPV 9) to 0.068 (HPV 5) (Appendix A). 

Lastly, RNA in situ hybridization (RNA ISH) for β-HPV genotype specific E6/E7 was performed on eleven (11/26, 42.3%) cases with sufficient tissue available on archival block. No signal was detected in any of the samples. Of the eight p16-negative cases with detectable hrHPV, hrHPV RNA ISH revealed one sample with detectable HPV mRNA (Appendix A).

### 3.4. Clinical Relevance

We examined the clinical relevance of the presence of various HPV types on disease prognosis. At the time of data cut-off, the median overall survival (OS) across p16-positive and -negative lesions was 69.7 months, and five (5/26, 19.2%) patients had died. There was no correlation between p16 or p53 status (normal vs. abnormal) and OS. Although not statistically significant, survival analysis by the presence of HPV types revealed a trend towards longer survival in patients who had no detectable HPV or beta HPV only vs. patients with detectable alpha HPV as a single or a co-infection with beta (Figure 3).

## 4. Discussion

In this study, we aimed to investigate the presence of various HPV types in vulvar squamous intraepithelial lesions. Apart from the well-recognized alpha HPV types, we found cutaneous-type HPV genotypes occurring as single infections or as co-infections with alpha HPVs in squamous intraepithelial lesions of the vulva. We further showed that high viral loads of beta HPVs could be detected in vulvar squamous intraepithelial lesions. The findings of this study should be considered to be hypothesis generating, and require validation in larger studies.

While β-HPV was present in p16-negative and -positive lesions alike in our study, the presence of β-HPV alone was associated with p16-negative lesions but not with conventional HSIL. We hypothesize that these p16-negative squamous intraepithelial lesions of the vulva may be similar to cutaneous squamous lesions such as actinic keratosis (AK), which have also been associated with cutaneous HPV types (e.g., β-HPV), which may be implicated in carcinogenic transformation by way of initiating the carcinogenic process but not contributing to its progression [32]. A recent meta-analysis showed a significant association of HPV 5, 8, 17, 20, 24, and 38 with the development of cSCC [32], which is congruent with the types identified in our intraepithelial lesions associated with vSCC. Our findings raise the possibility that the vulvar squamous lesions harbor mucosal- and cutaneous-type HPVs which may act in concert or with another carcinogenic agent (e.g., UV radiation) to induce carcinogenesis (Figure 4). On the one hand, even when associated with invasive squamous cell carcinoma, p16-negative squamous intraepithelial lesions investigated in the present study frequently had co-infections with β- and α-papillomavirus, suggesting that perhaps the types are able to initiate carcinogenesis without E6 or E7 integration into the host genome, as is classically described in high-risk HPV types [49]. On the other hand, this remains controversial with some authors suggesting these papillomaviruses may, instead, play a commensal role by activating CD8+ T cells for clearance of cells infected with β-HPV [50], thus necessitating additional preclinical studies. 

As part of this hypothesis-generating study, we further aimed to quantify the viral loads of selected beta HPV types in lesions with sufficient DNA content. The quantification of β-HPV DNA in the p16-negative intraepithelial lesions described herein showed near or more than one copy of β-HPV DNA per cell in the intraepithelial and invasive components of selected samples. Unfortunately, the assay sensitivity analysis revealed only two of the viral load quantification experiments had significant results, i.e., recorded viral loads above the LOD. Interestingly, the viral loads observed in the present study are in the range of the highest reported viral loads in actinic keratosis (one HPV DNA copy per <5 cell equivalents) [35]. The viral load observed in one of our samples (HPV 111) is comparable to viral loads observed in precancerous and cancerous skin tumors of patients with epidermodysplasia verruciformis (EV) ranging from 10 to more than 400 c/cell [51]. The highest viral loads were identified in a lesion harboring HPV 111 alone, in both the intraepithelial and invasive SCC. Since HPV 111 is a relatively newly described HPV genotype, limited information in the literature is available on the distribution, viral load, and carcinogenic potential. The available studies published in the literature on HPV 111 have reported its presence in AK and cSCC [52,53] as well as the anal canal in men who have sex with men [54]. In their phylogenetic analysis, Vasiljevič et al. [52] showed viral loads of HPV 111 in the range of 1/2000 cells in AK, and 1/300–6000 cells in basal cell carcinomas. 

Furthermore, the viral loads observed in our matched intraepithelial and invasive vulvar lesions were in line with the observations made by Weissenborn and colleagues in skin lesions, where they found much lower viral loads in non-melanoma skin cancer (NMSC) compared to AK. However, in our study, the viral loads of β-HPV detected in the invasive vulvar lesions appeared to be higher than those observed in NMSC, and they were more in the range of what Weissenborn et al. had found in AK [35]. While our findings are based on a very small set of samples and should be interpreted with caution, we believe this data warrant further investigation into the role of β-HPV in vulvar intraepithelial lesions and carcinomas, particularly those with HPV 111. Further studies should include quantification of HPV DNA. Based on our observations, a subset of p16-negative vulvar intraepithelial lesions may have abundant β-HPV; and therefore, might be associated with β-HPV infection as a potential cofactor. Additionally, these findings may be relevant for differentiation of ASP from true dVIN.

Additional carcinogens, such as the systemic effects of UV radiation or another unknown environmental or host factor, may act in synergy with β-HPV in the early stages of carcinogenesis [20,23,37]. Studies in murine models have suggested that β-HPVs tend to promote cellular proliferation in the presence of other stressors, such as UV radiation, thus facilitating the accumulation of DNA damage caused by UV radiation; however, this has never been studied in vSCC or sites with low direct UV exposure [37]. Mutations identified in association with p16-negative well-differentiated exophytic/verrucous vulvar lesions (*PIK3CA*, *NOTCH1*) overlap with those seen in preclinical models of β-HPV, and suggest that these lesions are biologically related, warranting further study [24,37]. In our study, none of the intraepithelial lesions from patients with immunosuppression (all categorized morphologically as HSIL, *n* = 3) had detectable β-HPV alone. One patient had a co-infection with α- and β-HPV types, and the other two patients only had α-HPV detected. Previously described risk factors such as EV, smoking, UV radiation, and immunosuppression associated with higher β-HPV prevalence and development of squamous cell carcinoma did not seem to be associated with β-HPV only in our cohort; however, our sample size was too small to draw any conclusions. 

Similarly, we did not observe any statistical significance between OS and p16 or p53 status, or the presence of different HPV genera. There is increasing recognition that p16-negative and/or p53-mutated vSCC have higher rates of recurrence and shorter OS compared to their p16-positive and/or p53wt counterparts [14,55,56]. The lack of statistical significance in our dataset is likely related to the small sample size and small number of events. Interestingly, the OS of patients with detectable beta HPV as a single infection was comparable to patients with alpha- and beta-independent tumors, whereas patients with detectable alpha HPV in their tumors (as single or co-infection) had poorer OS. The clinical implications of these observations remain unclear. Future studies should investigate the potential differences in clinical outcomes between patients harboring alpha single infections vs. mixed alpha/beta-infected lesions. Identifying the presence of beta co-infection as a potential prognostic factor may assist in tailoring patient follow-up and aggressiveness of treatment [40], especially considering the effects of radical surgical approaches on patients’ quality of life, sexual and urinary functions, and psychosocial well-being [57,58].

In addition, our data clearly demonstrate the discrepancy between HPV DNA positivity and p16 overexpression, since several cases initially grouped as p16 negative did, in fact, have detectable alpha HPV DNA and two of our p16-positive cases had no detectable alpha HPV DNA, which could reflect the high sensitivity of the Luminex assay to detect even a few rare cells harboring HPV infection. As we have seen in the subset of cases in which we performed qPCR, detectable HPV DNA does not necessarily correlate with significant viral load. Another group using the Luminex assay has reported similar observations with 50% of the HPV DNA positive vSCC cases showing no evidence of p16 overexpression [34]. While the synergistic model of UV radiation and β-HPV is mainly based on the transformative activities of E6 and E7 (mainly with relation to α-HPV types 10, 33, 53, and 54) leading to carcinogenesis in cSCC, there may be another HPV-driven pathway to carcinogenesis that could account for the p16-negative phenotype observed in many of the intraepithelial lesions [25]. 

The main limitations to our study are the limited sample size and the lack of sequencing data to define the molecular pathways associated with specific HPV types. In addition, the archival tissue samples used to investigate the β-HPV viral loads in matched intraepithelial and invasive blocks were from a single patient. Although significant efforts were made to select highly representative blocks with a well-defined intraepithelial or invasive component, some cases had focal intraepithelial lesions admixed/adjacent to invasive cancer, and we do not have assurance that we were indeed investigating the viral load on one component and not both. Although RNAISH for these β-HPV types was attempted, due to technical challenges and the lack of positive controls, we were unable to verify the location of the β-HPV in the tissue sections. Additional studies exploring the pathways that are altered in β-HPV associated lesions and co-infections with HR-HPV and the role of p53 in the transformation and carcinogenic initiation in vulvar dysplasia are needed. 

## 5. Conclusions

In conclusion, we identified an association between p16-negative squamous intraepithelial lesions with the presence of β-HPV types, suggesting an alternate etiologic pathway that may be involved in carcinogenic initiation, which is unique from high-risk HPV E6/E7 integration and which remains unknown in β-HPV. Furthermore, we present findings that may help to refine the HPV-independent category of vulvar intraepithelial lesions, which may be prognostically significant, and therefore improve our ability to identify these lesions. Based on our observations, further studies of β-HPV across the spectrum of precancerous and cancerous lesions of the vulva would be highly relevant.

## Figures and Tables

**Figure 1 viruses-15-01950-f001:**
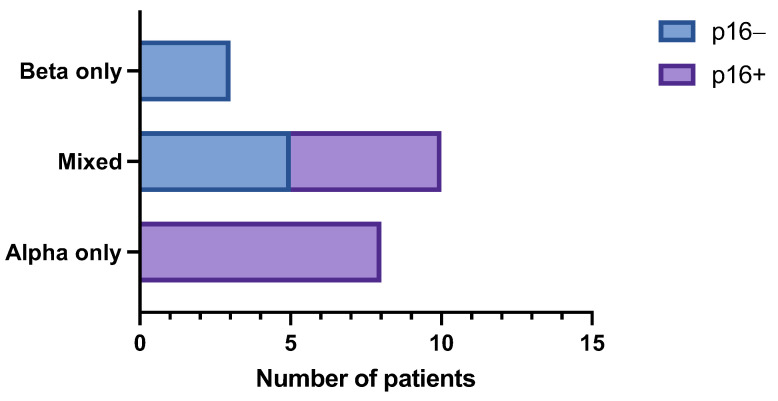
Figure depicting the number of samples with detectable human papillomavirus (HPV) DNA by Luminex technology relative to intraepithelial morphology.

**Figure 2 viruses-15-01950-f002:**
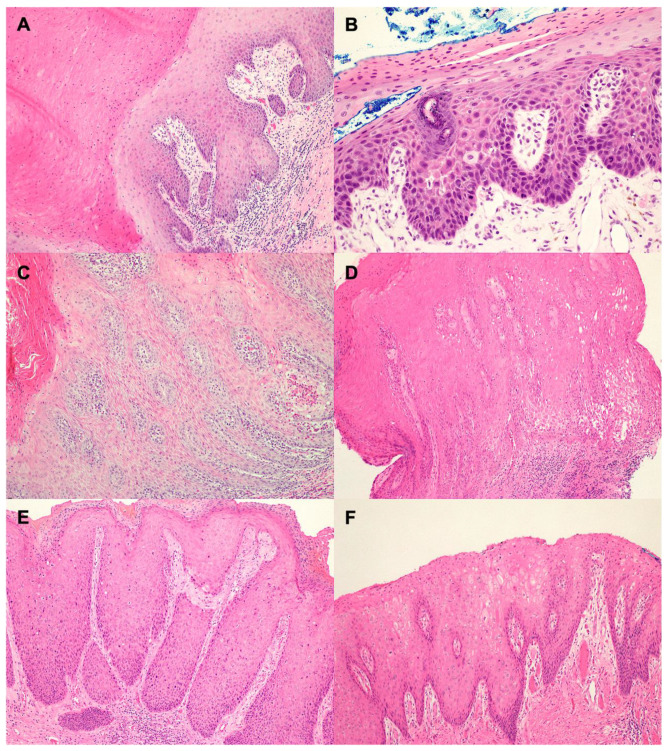
Hematoxylin and eosin stained sections of β-HPV-associated squamous lesions: (**A**) Case 5, β-HPV only intraepithelial lesion; (**B**) Case 44, β-HPV only intraepithelial lesion; (**C**) Case 28, β- and alpha (non-integrated) HPV; (**D**) Case 45, β- and alpha HPV; (**E**) Case 20, integrated HPV 16 with β- and gamma-HPV co-infections; (**F**) Case 24, HPV 16 with β- and gamma-HPV co-infections.

**Figure 3 viruses-15-01950-f003:**
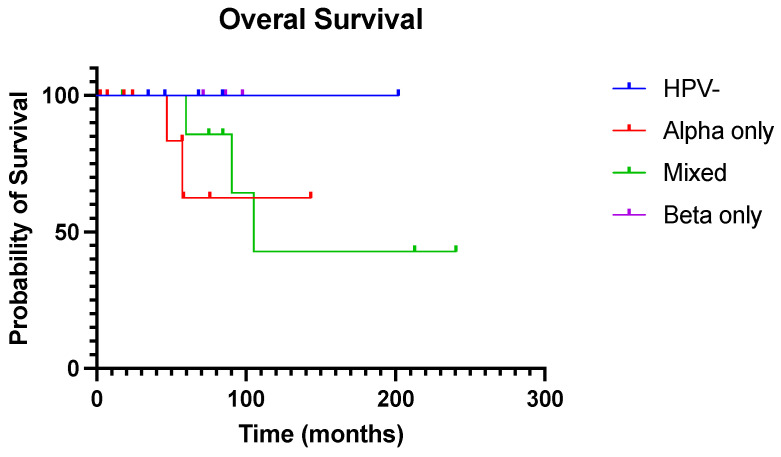
Overall survival (OS) relative to the presence of various HPV genera in the precursor vulvar intraepithelial lesions. Patients with detectable alpha HPV as a single infection or as a co-infection with beta HPV (mixed) have shorter OS (p = NS).

**Figure 4 viruses-15-01950-f004:**
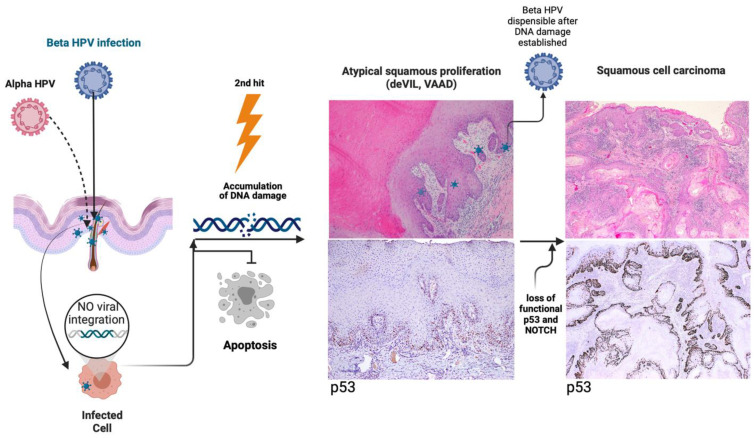
Proposed mechanism of action of β-HPV in the vulva. As represented in the schematic on the left, β-HPV infects cells, preferentially within hair follicles, without viral integration with or without alpha-HPV co-infection. There is accumulation of DNA damage via some other carcinogenic insult, which causes loss of functional p53 and/or alterations of the NOTCH signaling pathway, and ultimately squamous cell carcinoma, where beta HPV may or may not be detectable. The atypical squamous proliferation (ASP) shown in the panels in the middle (top, H&E with virus emblems representative of the positive HPV genotyping results, bottom: p53 immunohistochemistry) is a case from our series, with matched invasive carcinoma in the right panels (top, H&E and bottom, p53 staining). deVIL, differentiated exophytic vulvar intraepithelial lesions; VAAD, vulvar acanthosis with altered differentiation.

**Table 1 viruses-15-01950-t001:** Key patient, intraepithelial lesion, and invasive tumor characteristics. dVIN, differentiated vulvar intraepithelial neoplasia; HSIL, usual type high-grade squamous intraepithelial lesion; ASP, atypical squamous proliferation; VIN3, vulvar intraepithelial neoplasia grade 3; SCC, squamous cell carcinoma.

Characteristic	All Patients *n* = 26		HSIL ** *n* = 14		dVIN *n* = 3		ASP ** *n* = 9	
Mean age at diagnosis, years (range)	62.2	(31–88)	56.6	(31–80)	74.0	(65–87)	67.0	(40–88)
Smoking Status								
Current or former	13	(50.0%)	8	(57.1%)	1	(33.3%)	4	(44.4%)
Never	13	(50.0%)	6	(42.9%)	2	(66.7%)	5	(55.6%)
p16, *n* (%)								
Positive	14	(53.8%)	14	(100%)	0	(0)	0	(0)
Negative	12	(46.2%)	0	(0)	3	(100%)	9	(100%)
p53, *n* (%)								
Overexpressed basal	7	(26.9%)	2	(14.3%)	0	(0)	5	(55.6%)
Basal sparing	10	(38.5%)	10	(71.4%)	0	(0)	0	(0)
Mutant overexpression	4	(15.4%)	1	(7.1%)	3	(100%)	0	(0)
Wild type	5	(19.2%)	1	(7.1%)	0	(0)	4	(44.4%)
SCC Subtype, *n* (%)								
Keratinizing	17	(65.4%)	8	(57.1%)	3	(100%)	6	(66.6%)
Non-keratinizing	3	(11.5%)	3	(21.4%)	0	(0)	0	(0)
Verrucous	4	(15.4%)	1	(7.1%)	0	(0)	3	(33.3%)
Missing	1	(3.8)	1	(7.1%)	0	(0)	0	(0)
SCC Invasion Type								
Infiltrative	20	(76.9%)	12	(85.7%)	2	(66.7%)	6	(66.7%)
Pushing	5	(19.2%)	2	(14.3%)	1	(33.3%)	2	(22.2%)
Infiltrative/pushing	1	(3.8%)	0	(0)	0	(0)	1	(11.1%)
Multifocal, *n* (%)								
Yes	5	(19.2%)	3	(21.4%)	1	(33.3%)	1	(11.1%)
No	21	(80.1%)	11	(78.6%)	2	(66.7%)	8	(88.9%)
Immunosuppression, *n* (%)								
Yes	3	(11.5%)	3	(21.4%)	0	(0)	0	(0)
No	23	(88.5%)	11	(47.8%)	3	(100%)	9	(100%)
Initial Intraepithelial Diagnosis, *n* (%)								
ASP	2	(7.6%)	0	(0)	0	(0)	1	(11.1%)
dVIN	9	(34.6%)	1	(7.1%)	3	(100%)	5	(55.6%)
HSIL, usual type	12	(46.2%)	11	(78.6%)	0	(0)	1	(11.1%)
VIN3, mixed usual/differentiated	1	(3.8%)	0	(0)	0	(0)	1	(11.1%)
Verrucous squamous lesion	1	(3.8%)	1	(7.1%)	0	(0)	0	(0)
No definitive precursor diagnosis *	1	(3.8%)	0	(0)	0	(0)	1	(11.1%)

* reclassified as ASP; ** one patient with 2 cases with comparable tumor characteristics.

**Table 2 viruses-15-01950-t002:** Summary of human papillomavirus (HPV) types categorized by p16 status (*n* = 26 cases from 26 patients). N, negative and P, positive.

p16 Status	Case	HPV Type	
		β	High-Risk α
N	3		16
N	5	111	
N	6	105	16
N	11	5	31, 51
N	15	5, 23	
N	28	111	31, 51
N	42		
N	45	9	16
N	57	110	16
N	7		
N	47		
N	10	24	
P	20	75, 145	16
P	21		16, 31
P	22		
P	24	36, 25	16, 39
P	26		16, 31
P	31		18
P	32		16
P	34		
P	36		16, 31
P	40		16
P	49		31, 56
P	50	9	16
P	51		16
P	56		16

**Table 3 viruses-15-01950-t003:** Cutaneous (beta) human papillomavirus (HPV) viral loads as determined by qPCR in vulvar intraepithelial lesions with sufficient DNA isolated from separately intraepithelial and invasive cancer formalin-fixed paraffin-embedded blocks. Three cases had sufficient DNA available from both intraepithelial and invasive carcinoma-containing blocks. dVIN, differentiated vulvar intraepithelial neoplasia; ASP, atypical squamous proliferation; HSIL, usual type high-grade squamous intraepithelial lesion; LOD, limit of detection; NA, not applicable.

Intraepithelial Morphology	HPV Genotype	Assay LOD	Intraepithelial Viral Load *	Invasive Viral Load *	Patient Number	Co-Infection with α-HPV
HSIL	HPV 9	0.19	0.94	NA	50	Yes, HPV 16
HSIL	HPV 24	1.21	0.23	0.16	20	Yes, HPV 16
ASP	HPV 111	2.72	153.60	105.62	5	No
ASP	HPV 5	3.01	0.05	NA	15	No
ASP	HPV 5	3.01	1.64	0.25	5	No

* Presented as copies/cell.

## Data Availability

All data generated as part of this study are available in Appendix A.

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
