# Peer review of "Betapapillomaviruses in p16-Negative Vulvar Intraepithelial Lesions Associated with Squamous Cell Carcinoma"

_viruses, 2023, doi:10.3390/v15091950_

Round 1
Reviewer 1 Report
Thank you for allowing me to review this excellent piece of work. The authors present a relatively novel analysis of vulvar lesions. They use contemporary language to describe the vulvar lesions. Their presentation is logical and informative. The tissue analysis is rich and relatively cutting edge. The methods are so clearly presented, they could be replicated and given that no one institution has massive numbers, a systematic review of similar works could add to the finding of this work in the future. The paper is rich in content. Limitations are clearly articulated.
Author Response
We would like to thank the reviewer for taking the time to review our manuscript. We appreciate your supportive critique.
Reviewer 2 Report
Dear authors. I believe this article is very well written and also important for better understanding of HPV in connection VIN and vulvar carcinoma.
I have some minor suggestions/questions to clarify:
- in table 1, ASP is explained twice
- While the article mentions associations between HPV types and clinical outcomes, it could benefit from a more in-depth discussion of the clinical implications of these findings. How might the presence of β-HPV affect diagnosis, treatment, or prognosis for patients with vulvar intraepithelial lesions? Do you think such discussion is possible?
- So if I understand correctly you would describe why viral load was higher in lesions than in carcinoma with the fact that this may be due to a role in the initiation of carcinogenesis, but not in its progression? That was described in line 78 but less in the discussion.
Author Response
We thank the reviewer for taking the time to review our manuscript and for their thoughtful suggestions on how to improve it. We have made the following changes to the manuscript:
- The duplicated explanation of ASP has been removed from Table 1.
- We would like to thank the reviewer for pointing out the potential clinical implications of our findings. Our cohort was designed with an emphasis on p16 negative verrucous-like lesions of the vulva and we did not include many dVINs or cancers to be able to make meaningful conclusions related to prognosis. We address the potential for improving the diagnosis of vulvar precancerous lesions in line 451-2; i.e., the relevance to differentiate ASP from true DVIN lesions. In addition, we have added a brief discussion on the potential impacts of beta HPV presence on clinical management (lines 478-484). Our findings suggest patients with alpha and beta coinfections and alpha single infections do worse compared to beta-only or HPV-negative patients; however, our sample size is too small to draw meaningful conclusions. We propose future studies should focus on elucidating this trend further; particularly whether there are any differences between alpha only and alpha/beta mixed lesions.
- Yes, we hypothesize that beta HPV plays a role in disease initiation, but not in its progression and invasion, similarly to cutaneous squamous intraepithelial lesions. That is explained in the Introduction (lines 77-80) and in the Discussion (lines 395-397). We've made a minor change to line 80 and line 396 to improve clarity.
Reviewer 3 Report
This is a rigorous work in an interesting and exciting niche. Great work authors.
I would only address some points for publication
IN introduction I would expand discussion about vulvar cancer trends..(e.g. Preti M, Bucchi L, Micheletti L, et al. Four-decade trends in lymph node status of patients with vulvar squamous cell carcinoma in northern Italy. Sci Rep. 2021;11(1):5661. doi:10.1038/s41598-021-85030-x)
Regarding pre invasive squamous lesions i would suggest citing this multisociety recent statement: Preti M, Joura E, Vieira-Baptista P, et al. The European Society of Gynaecological Oncology (ESGO), the International Society for the Study of Vulvovaginal Disease (ISSVD), the European College for the Study of Vulval Disease (ECSVD) and the European Federation for Colposcopy (EFC) consensus statements on pre-invasive vulvar lesions. Int J Gynecol Cancer. 2022;32(7):830-845. doi:10.1136/ijgc-2021-003262
Author Response
We thank the reviewer for taking the time to review our manuscript and for their thoughtful suggestions on how to improve it. We have made the following changes to the manuscript:
- We have added a brief discussion on vulvar cancer trends to the introduction (91-92) to illustrate the need for early detection and improved understanding of this disease to combat late-stage diagnosis.
- The ESGO multisociety statement has been cited in line 484 where we have added a paragraph discussing the potential clinical implications of our findings; i.e. tailoring patient follow-up and aggressiveness of treatment relative to risk of recurrence.